# Low QRS Voltage in Limb Leads Indicates Accompanying Precordial Voltage Attenuation Resulting in Underestimation of Left Ventricular Hypertrophy

**DOI:** 10.3390/ijerph182412867

**Published:** 2021-12-07

**Authors:** Hye-Bin Gwag, Su-Hyun Lee, Hyeon-Jun Kim, June-Soo Kim, Young-Keun On, Seung-Jung Park, Kyoung-Min Park

**Affiliations:** 1Division of Cardiology, Department of Internal Medicine, Samsung Changwon Hospital, Sungkyunkwan University School of Medicine, Changwon 51353, Korea; tgfbk@naver.com; 2Department of Internal Medicine, Samsung Changwon Hospital, Sungkyunkwan University School of Medicine, Changwon 51353, Korea; leesu9003@gmail.com; 3Ewha Womans University Medical Center, Mokdong Hospital, Seoul 07985, Korea; 70453@eumc.ac.kr; 4Division of Cardiology, Department of Internal Medicine, Samsung Medical Center, Sungkyunkwan University School of Medicine, Seoul 06351, Korea; js58.kim@samsung.com (J.-S.K.); yk.on@samsung.com (Y.-K.O.); orthovics@gmail.com (S.-J.P.)

**Keywords:** electrocardiogram, limb lead, QRS, voltage, left ventricular hypertrophy

## Abstract

Low QRS voltage (LQRSV) in electrocardiography (ECG) often occurs in limb leads without apparent cause. However, its clinical significance is obscure in healthy populations. We reviewed patients aged over 60 who were scheduled for non-cardiac surgery in two hospitals. Patients underwent pre-operative ECG, echocardiography, pulmonary function test, and chest X-ray. Patients with LQRSV isolated to limb leads and patients without LQRSV were selected from separate hospitals. Among the 9832 patients screened in one hospital, 292 (3.0%) showed LQRSV in limb leads. One-hundred and ninety-four without LQRSV were selected as the control from the 216 patients screened at the other hospital. For primary analysis, patients with structural heart disease or classic etiologies of LQRSV were excluded. Patients with LQRSV had a higher proportion of male and a greater body mass index. Precordial QRS voltages were smaller, whereas left ventricular mass index and the prevalence of echocardiographic left ventricular hypertrophy (LVH) was higher in patients with LQRSV than in those without. Consequentially, diagnostic performance of precordial voltage criteria for LVH was particularly poor in patients with LQRSV in limb leads. LQRSV in limb leads frequently occurs without apparent etiologies. ECG voltage criteria may underestimate LVH in a relatively healthy population with LQRSV in limb leads.

## 1. Introduction

Low QRS voltage (LQRSV) in surface electrocardiography (ECG) can be observed in various populations not infrequently [1,2,3,4,5,6]. Discordant QRS voltage refers to when LQRSV is limited to either limb or precordial leads. A small number of studies demonstrated that dilated cardiomyopathy and so-called classic etiologies including infiltrative cardiomyopathy, pericardial thickening or effusion, pulmonary diseases, or obesity, were associated with discordant QRS voltage [2,6]. These etiologies seem theoretically plausible because LQRSV can be a consequence of impaired voltage generation of ventricular myocardium and/or signal attenuation during electrical conduction [1,7]. However, none of the previous studies could explain why LQRSV is present only in limb or precordial leads, but not in both [2,6]. Thus, we sought to find the etiologies of LQRSV limited to limb leads, a frequent form of discordant QRS voltage, and to investigate clinical significance of unexplained LQRSV in this study. We included patients undergoing standardized pre-operative examinations for non-cardiac surgery because they represent a relatively healthy population in which underlying cardiomyopathy or classic etiologies for discordant QRS voltage could be ruled out.

## 2. Materials and Methods

We screened consecutive patients aged over 60 who underwent preoperative risk evaluation for non-cardiac surgery between January 2016 and December 2016 in two tertiary university hospitals. All patients routinely underwent ECG and echocardiography for cardiac screening and chest radiography, and pulmonary function for preoperative pulmonary risk evaluation. All examinations were performed within a week. We categorized patients according to LQRSV, which was defined as QRS complex voltages ≤0.5 mV in all limb leads or ≤1.0 mV in all precordial leads [2,8]. We identified patients with LQRSV isolated in limb leads from one hospital (LQRSV group) and selected all patients without LQRSV in both limb and precordial leads as a control group among corresponding patients from the other hospital (Figure 1). Demographics data were collected through careful reviews of electronic medical records. Standard postero-anterior chest radiography was used for the diagnosis of pulmonary hyperinflation, emphysema, and pleural effusion. Large to massive pleural effusion was defined to occur when it occupied more than two thirds of the hemithorax [9,10]. The diagnosis of obstructive pulmonary disease was confirmed if the FEV1/FVC ratio was less than 70% on pulmonary function test [11]. Structural heart disease included LV systolic dysfunction, ischemic, or non-ischemic cardiomyopathy, and valvular heart disease. Moderate or large-sized pericardial effusions, pericardial thickening or fat, infiltrative cardiomyopathy, pulmonary hyperinflation, emphysema, obstructive lung disease, large and massive pleural effusions, and body mass index ≥ 30 kg/m^2^ were defined as classic etiologies for LQRSV. The Institutional Review Board at the two hospitals approved the study protocol, and informed consent was waived by the board. The study protocol conformed to the ethical guidelines of the 1975 Declaration of Helsinki.

All 12-lead ECGs were recorded at a paper speed of 25 mm/s using a Philips PageWriter TC30 cardiograph device (Philips Healthcare, Andover, MA, USA) with 0.05–150 Hz bandpass filters. A general cardiologist and an electrophysiologist reviewed ECGs in a blind fashion and used a digital caliper under at least 2-fold magnification for manual measurement of ECG parameters. Electrocardiographic parameters included baseline rhythm, QRS voltage, QRS duration, precordial R/S transition, frontal axis of QRS complex, PR interval, and corrected QT interval. A normal precordial R/S transition zone was defined as V3 or V4, and a normal QRS axis was defined as −30° to 90° [12]. The tallest R and deepest S wave voltages in any precordial lead were also measured in each patient (Figure 2). Electrocardiographic left ventricular hypertrophy (LVH) criteria were defined as follows: (1) tallest R wave > 2.5 mV, (2) deepest S > 2.3 mV, or (3) tallest R + deepest S > 3.5 mV by adopting the existing ECG criteria for LVH [13,14,15,16,17].

Echocardiographic measurements included LV ejection fraction (LVEF), LV end-diastolic and -systolic diameters, LV mass, and amount of pericardial effusion. The amount of effusion was graded as moderate when echo-free space in the diastole was >10 mm [18]. Ventricular dilatation was defined as end-diastolic diameter >56 mm and >30 mm, respectively for left and right ventricles [2]. LV mass index (LVMI, g/m^2^) was calculated as LV mass divided by body surface area. Echocardiographic LVH was defined as LVMI ≥96 g/m^2^ in women and ≥116 g/m^2^ in men, according to the 2015 American Society of Echocardiography guidelines [19]. Valvular heart disease was defined as valvular stenosis or regurgitation of moderate or greater degree.

Categorical variables were shown as number and percentage and were compared using the χ2 test. Continuous variables were shown as means with the standard deviation and compared using the Student *t*-test. *p* values less than 0.05 were considered statistically significant. Patients without structural heart disease or known classic etiologies of LQRSV [2] were included for primary analysis. Logistic regression models were used to determine the association between other co-variates and low QRS voltage limited to limb leads. Variables with *p*-value < 0.1 in univariate analysis were included in multivariate analysis. Receiver operating characteristic (ROC) analysis was performed to evaluate diagnostic ability of ECG voltage criteria for the detection of echocardiographic LVH. The Clopper–Pearson procedure was used to calculate 95% confidence intervals for estimates of sensitivity and specificity of ECG LVH criteria. The McNemar test was used to assess the agreement between electrocardiographic and echocardiographic LVH. All statistical analyses were performed using SPSS Statistics Version 23 (IBM Corporation, Armonk, NY, USA).

## 3. Results

### 3.1. Patient Categorization and Baseline Characteristics

A total of 10,048 patients (9832 and 216 patients from respective hospitals) were screened and categorized according to the presence of LQRSV in precordial and/or limb leads. In both hospitals, most patients showed no LQRSV in any lead (97.0% and 89.8%, respectively), while those with LQRSV only in limb leads accounted for the second highest proportion (3.0% and 10.2%, respectively) (Figure 1). Baseline characteristics were compared between 292 patients with LQRSV in limb leads and 194 control patients without LQRSV (Appendix A). The LQRSV in the limb leads group had more men and a higher body mass index compared to the control group. Ventricular dilatation was less frequent, whereas the classic etiologies of LQRSV were more frequent in patients with LQRSV in limb leads than in the control patients.

### 3.2. Analysis of Patients without Structural Heart Disease or Classic Etiologies of LQRSV

For primary analysis, we excluded patients with structural heart disease and known classic etiologies of LQRSV. Table 1 shows a comparison of baseline data between the groups with LQRSV in limb leads and the control group. Patients with LQRSV in limb leads (n = 236) showed a higher prevalence of men and had a greater body mass index compared to those without LQRSV (n = 167). There were also significant differences in echocardiographic parameters between the two groups. The LQRSV group had smaller LV end-systolic dimension (28.2 ± 3.4 vs. 30.3 ± 4.2, *p* < 0.001), higher LV mass index (100.9 ± 22.9 vs. 93.5 ± 18.4, *p* = 0.001), and more echocardiographic LVH (33.5% vs. 24.0%, *p* = 0.04) than the control group. In a comparison of ECG parameters, baseline atrial fibrillation, late R/S transition, and left QRS axis deviation were more frequent in patients with the LQRSV compared to those without. Unlike LVMI, the tallest R voltage and the tallest R + deepest S voltages in precordial leads were smaller in the LQRSV group (13.5 ± 4.3 vs. 18.5 ± 6.7, *p* < 0.001 and 25.7 ± 6.2 vs. 30.4 ± 8.7, *p* < 0.001, respectively). In multivariate logistic regression analysis, we identified that male, smaller LV end-systolic dimension, greater LV mass index, baseline atrial fibrillation, smaller pre-cordial tallest R voltage, and shorter QT interval were significantly associated with low QRS voltage limited to limb leads (Table 2).

### 3.3. Diagnostic Performance of Precordial Voltages to Predict Echocardiographic LVH

In the control group, the two ECG criteria for LVH (tallest R wave > 2.5 mV and tallest R + deepest S wave > 3.5 mV) showed area under the curve (AUC) values of 0.63 and 0.60 (95% confidential interval 0.54–0.72, *p* = 0.01 and 0.51–0.70, *p* = 0.05, respectively), while the other criteria (deepest S wave > 2.3 mV) did not show significant diagnostic ability for LVH (*p* = 0.59). In contrast, all three criteria were non-diagnostic in the LQRSV group (Figure 3). The overall sensitivities of the threee criteria were not only low, but especially low in the LQRSV group, so that sensitivities of the two criteria using the tallest R and/or deepest S wave voltages were 0%. The McNemar test also showed poor agreement between the electrocardiographic and echocardiographic LVH in the LQRSV group (Table 3).

## 4. Discussion

The surface 12-lead ECG records electrical signals traveling from the heart to the body surface electrodes. It is generally known that LQRSV on ECG is associated with conditions either impairing voltage generation of the myocardium or altering signal transmission from the heart to the skin electrodes. LQRSV can, however, be encountered not infrequently without apparent cause [1]. Occasionally, LQRSV occurs only in limb leads or only in precordial leads and is then known as discordant QRS voltage. The standard surface ECG consists of two different recording modes (unipolar or bi-polar). Three standard limb leads I, II, and III have two dedicated electrodes, but the precordial leads have a single electrode and a constructed reference known as the Wilson’s central terminal. The lack of correlation between the voltages of different leads has been noted [2,6,20], and efforts have been made to explain the cause of discordant QRS voltage. One previous study sought clinical conditions causing LQRSV isolated to the limb leads, which is far more frequently observed than that isolated to precordial leads [2]. In this study, half the patients had well-known conditions for diffuse LQRSV, so-called classic etiologies, which include moderate or large-sized pericardial effusions, pericardial thickening or fat, pulmonary hyperinflation or emphysema, infiltrative cardiomyopathy, large and massive pleural effusions, and body mass index ≥ 30 kg/m^2^. Among the remaining half of patients, 63% had dilated LV with reduced LVEF. This can be understood in the same context as other studies that showed voltage in precordial leads correlated with LV dimensions, while that of limb leads had an inverse relationship with peripheral edema in heart failure patients, resulting in voltage discrepancy between limb and precordial leads [20,21]. A considerable number of patients (18%) in the study still, however, had unexplained LQRSV.

Regardless, clinical significance of unexplained LQRSV in limb leads is uncertain to date, especially in healthy populations. This is one reason why we specifically studied a population healthy enough to undergo non-cardiac surgery and who underwent pre-operative routine examinations in which the classic etiologies for LQRSV or LV size and function could be identified. Our study population showed a much higher proportion of unexplained LQRSV compared with the previous study (81.5% vs. 18%). In our study, only 13.7% of patients with LQRSV in limb leads had classic etiologies for LQRSV and five patients (1.7%) had dilated LV dimension, while the mean LVEF was 54.8%. This discrepancy can possibly be a result of different study populations and numbers. The previous study screened patients during routine ECG reading (n = 100), while we included patients aged over 60 who were scheduled for non-cardiac surgery (n = 292). Furthermore, the strength of our study, apart from the previous one, is that we compared clinical, electrocardiographic, and echocardiographic data of LQRSV patients to those of the control patients. There are two reasons why we selected control patients from the separate hospitals instead of using patients from the very hospital from which LQRSV patients were screened. First, it is practically difficult and time-consuming to measure plenty of 12-lead ECG parameters of all patients, which numbered over 9000. It also causes disproportionate patient numbers between two patient groups. We could consider a patient sampling method to overcome these problems, but instead, we decided to choose a simple way to include all patients from the second hospital following the same pre-operative screening protocol, albeit smaller in volume. In this regard, there can be possible differences in the confounding baseline characteristics between two patient groups.

As our primary interest was the clinical significance of unexplained LQRSV in the ‘healthy’ population, patients with structural heart disease or classic etiology of LQRSV were excluded from primary analysis. Interestingly, patients with LQRSV had smaller LV end-systolic diameter and higher LVMI as well as higher prevalence of late R/S transition and left axis deviation compared to the control patients, all of which are echocardiographic or electrocardiographic results indicating LVH. These results, except for significantly lower precordial voltages in patients with LQRSV in limb leads than the control, were consistent with the higher prevalence of LVH in patients with LQRSV. In the ROC curve analysis, usual voltage criteria cut-off values for LVH had no diagnostic ability in the LQRSV group. Not only were the sensitivities of the ECG LVH criteria in the LQRSV group lower than those in the control group, but the values were extremely low (0–8.4%), even given that the sensitivities of the existing voltage criteria are generally known to be low (usually less than 50%) [22]. Thus, it is necessary to bear in mind that the role of ECG voltage criteria to detect LVH is highly limited in patients with LQRSV isolated to limb leads. Taken together, a more plausible explanation for this result could be that LQRSV in limb leads implies proportional attenuation of precordial voltage rather than there being a specific etiology to reduce limb lead QRS voltage only, although we could not identify the cause for overall ECG voltage attenuation. It is known that LVH detected by 12-lead ECG has predictive value for cardiovascular morbidity and mortality, and regression of LVH on ECG is associated with cardiovascular risk reduction [23,24,25]. The importance of LVH detection has been based on those relevant studies. Surface ECG is a simple and reproducible tool for LVH screening, but low sensitivity of ECG LVH criteria has been a main limitation. Thus, a recent study even suggested a combination of two different ECG criteria for better risk prediction [26]. Considering that a patient can miss an opportunity to prevent LVH-related morbidity and mortality by underestimation of LVH, it is meaningful to investigate whether scoring systems with non-voltage components such as the Romhilt–Estes or Perugia scores can improve sensitivity or diagnostic performance of ECG to detect LVH in patients with LQRSV in limb leads [27,28,29].

In summary, LQRSV in limb leads can occur without classic etiologies for diffuse LQRSV or structural heart disease in a relatively healthy population. The ECG voltage criteria for LVH have little diagnostic performance in that population. It seems reasonable to regard LQRSV in limb leads as an indicator of accompanying attenuation of precordial voltages rather than as a consequence of specific etiology with clinical significance.

This study has several limitations. First, our results cannot be generalized to other populations with different ages or clinical situations. We collected patients for each study group from two different institutions. Even though both are tertiary university hospitals following the same protocol for pre-operative screening, there is a difference in patient and bed numbers due to regional disparity. There could be differences in the confounding baseline characteristics between two patient groups. Second, electrocardiographic LVH was defined by the tallest R or deepest S wave amplitudes instead of by R or S wave voltage of a certain lead as in conventional precordial voltage criteria. However, it is known that position changes of ECG electrodes resulting in day-to-day variation of ECG voltage can reduce reproducibility of the criteria using any fixed lead voltage, which implies that maximum voltage in any lead is more sensitive in detecting LVH [30,31,32]. Third, clinical outcomes of patients with LQRSV in limb leads were not evaluated in this study. It is worth further study considering that several studies have reported that LQRSV was associated with mortality in patients with cardiovascular diseases and even in disease-free populations [4,8,33].

## 5. Conclusions

In a relatively healthy population, more than 80% of LQRSV isolated to limb leads could not be explained by well-known causes. LQRSV in limb leads suggests accompanying precordial voltage attenuation and limits the role of ECG voltage criteria for LVH as a consequence in populations without structural heart disease or general causes for LQRSV. In this population, ECG criteria with non-voltage components seem to be appropriate for diagnosis of LVH.

## Figures and Tables

**Figure 1 ijerph-18-12867-f001:**
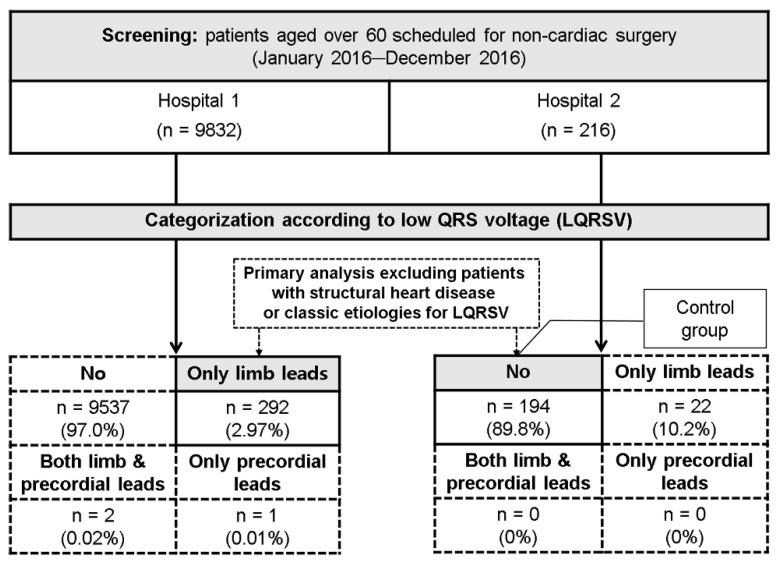
Study population.

**Figure 2 ijerph-18-12867-f002:**
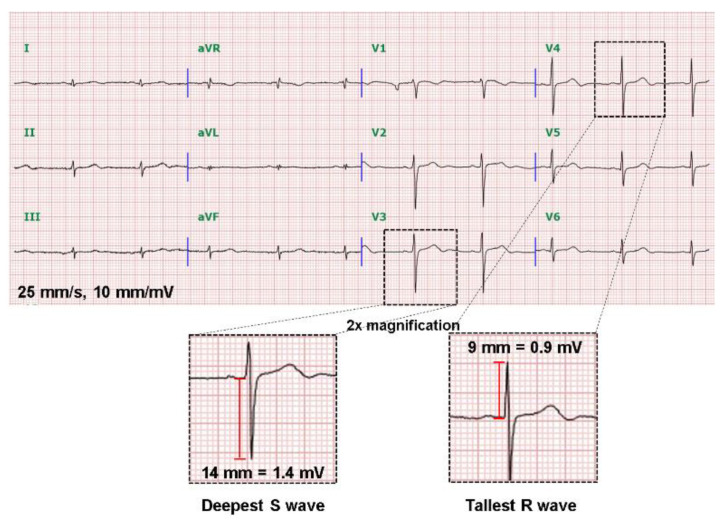
A 12-lead electrocardiogram showing low QRS voltage isolated in limb leads. The QRS voltages were ≤0.5 mV in all limb leads. The tallest R and the deepest S voltages were measured under 2-fold magnification.

**Figure 3 ijerph-18-12867-f003:**
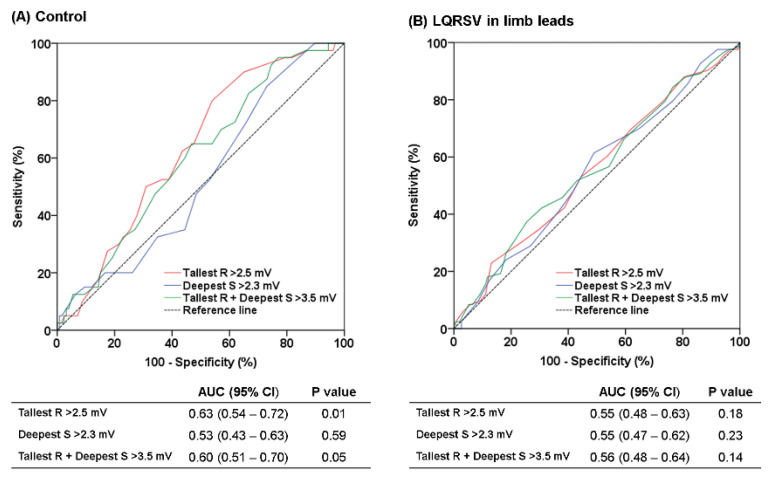
Receiver operating characteristic (ROC) curve with three precordial voltage criteria for detecting echocardiographic left ventricular hypertrophy in control patients without low QRS voltage (LQRSV) (**A**) and patients with LQRSV in limb leads (**B**). AUC, area under the curve; CI, confidence interval.

**Table 1 ijerph-18-12867-t001:** Baseline data in patients without structural heart disease and known classic etiologies of low QRS voltage (LQRSV).

	LQRSV in Limb Leads (n = 236)	Control (n = 167)	*p* Value
**Demographic data**			
Age (year)	71.1 ± 6.7	71.3 ± 6.1	0.78
Male	142 (60.2)	80 (47.9)	0.02
Height (cm)	160.5 ± 9.2	159.2 ± 8.7	0.15
Weight (kg)	60.9 ± 9.9	58.8 ± 9.4	0.04
BMI (kg/m^2^)	23.5 ± 2.7	23.2 ± 2.8	0.18
Hypertension	115 (48.7)	82 (49.1)	0.94
Diabetes	65 (27.5)	40 (24.0)	0.42
**Echocardiographic parameters**			
LV ejection fraction (%)	64.9 ± 5.7	65.3 ± 6.2	0.50
LV end-diastolic dimension (mm)	47.2 ± 4.2	47.5 ± 5.2	0.71
LV end-systolic dimension (mm)	28.2 ± 3.4	30.3 ± 4.2	<0.001
LV mass index (g/m^2^)	100.9 ± 22.9	93.5 ± 18.4	0.001
Echocardiographic LVH ^1^	79 (33.5)	40 (24.0)	0.04
**Electrocardiographic parameters**			
Baseline rhythm			
Sinus	215 (91.1)	160 (96.4)	0.04
Atrial fibrillation	21 (8.9)	6 (3.6)	0.04
QRS duration (ms)	88.1 ± 9.3	89.6 ± 12.5	0.16
Precordial R/S transition zone			
Normal	163 (69.1)	128 (77.1)	0.08
Early (V1 or V2)	12 (5.1)	19 (11.4)	0.02
Late (V5 or V6)	58 (24.6)	19 (11.4)	0.001
Frontal QRS axis	27.0 ± 41.5	38.0 ± 31.7	0.004
Normal (−30–90°)	207 (87.7)	160 (96.4)	0.002
Left axis deviation	18 (7.6)	3 (1.8)	0.01
Right axis deviation	11 (4.7)	3 (1.8)	0.12
Precordial voltages (mm) †			
Tallest R wave	13.5 ± 4.3	18.5 ± 6.7	<0.001
Deepest S wave	12.2 ± 3.9	11.8 ± 4.3	0.33
Tallest R+ deepest S wave	25.7 ± 6.2	30.4 ± 8.7	<0.001
PR interval (ms)	173.0 ± 24.3	170.4 ± 22.0	0.29
Corrected QT interval (ms)	430.8 ± 26.0	439.5 ± 27.0	0.001

Values are presented as the mean ± SD or number (percentage). ^1^ Defined as left ventricular mass index ≥ 96 g/m^2^ in women and ≥116 g/m^2^ in men. BMI, body mass index; LV, left ventricular; LVH, left ventricular hypertrophy. †Each 10 mm equals 1 mV.

**Table 2 ijerph-18-12867-t002:** Unexplained low QRS voltage (LQRSV) limited to limb leads and associated factors by logistic regression.

	Univariate Analysis		Multivariate Analysis	
	Odds Ratio (95% CI)	*p* Value	Odds Ratio (95% CI)	*p* Value
**Demographic data**				
Age (year)	0.996 (0.966–1.027)	0.783		
Male	1.643 (1.101–2.451)	0.015	2.714 (1.444–5.101)	0.002
Height (cm)	1.016 (0.994–1.039)	0.15		
Weight (kg)	1.022 (1.001–1.044)	0.037	1.010 (0.977–1.045)	0.558
BMI (kg/m^2^)	1.052 (0.977–1.132)	0.177		
Hypertension	0.985 (0.663–1.465)	0.941		
Diabetes	1.207 (0.765–1.904)	0.419		
**Echocardiographic parameters**				
LV ejection fraction (%)	0.989 (0.956–1.022)	0.501		
LV end-diastolic dimension (mm)	1.008 (0.966–1.052)	0.714		
LV end-systolic dimension (mm)	0.864 (0.817–0.914)	<0.001	0.713 (0.650–0.781)	<0.001
LV mass index (g/m^2^)	1.017 (1.007–1.027)	0.001	1.058 (1.040–1.076)	<0.001
**Electrocardiographic parameters**				
Baseline atrial fibrillation	2.605 (1.028–6.602)	0.044	4.584 (1.372–15.316)	0.013
QRS duration (ms)	0.987 (0.969–1.006)	0.167		
Late R/S transition	2.521 (1.437–4.423)	0.001	1.046 (0.492–2.223)	0.98
Left axis deviation	4.514 (1.308–15.581)	0.017	1.504 (0.368–6.152)	0.57
Precordial tallest R voltages (mm)	0.836 (0.798–0.876)	<0.001	0.783 (0.734–0.837)	<0.001
PR interval (ms)	1.005 (0.996–1.014)	0.29		
Corrected QT interval (ms)	0.988 (0.980–0.995)	0.002	0.988 (0.978–0.998)	0.017

CI, confidence interval.

**Table 3 ijerph-18-12867-t003:** Diagnostic performance of precordial voltages to detect echocardiographic left ventricular hyper-trophy. ^1^
*p*-value < 0.05 indicates lack of agreement. LR+, positive likelihood ratio; LR−, negative likelihood ratio. Each 10 mm equals 1 mV.

	Sensitivity (%)	Specificity (%)	LR+	LR−	McNemar Test ^1^
**LQRSV in limb leads**					
Tallest R > 25 mm 95% confidence interval	0.0 (0/83 × 100) [0.0–4.4]	100 (153/153 × 100) [97.6–100]	Not applicable	1.00	Not applicable
Deepest S > 23 mm 95% confidence interval	0.0 (0/83 × 100) [0.0–4.4]	98.7 (151/153 × 100) [95.4–99.8]	0.00	1.01	<0.001
Tallest R + deepest S > 35 mm 95% confidence interval	8.4 (7/83 × 100) [3.5–16.6]	93.5 (0/83 × 100) [88.3–96.8]	1.29	0.98	<0.001
**Control**					
Tallest R > 25 mm 95% confidence interval	17.5 (7/40 × 100) [7.3–32.8]	85.0 (108/127 × 100) [77.6–90.8]	1.17	0.97	0.07
Deepest S > 23 mm 95% confidence interval	2.5 (1/40 × 100) [0.1–13.2]	99.2 (125/126 × 100) [95.7–99.9]	3.13	0.98	<0.001
Tallest R + deepest S > 35 mm 95% confidence interval	32.5 (13/40 × 100) [18.6–49.1]	77.0 (97/126 × 100) [68.7–84.0]	1.41	0.88	0.89

## Data Availability

The data presented in this study are available in the Appendix A.

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
