# Peer review of "Low QRS Voltage in Limb Leads Indicates Accompanying Precordial Voltage Attenuation Resulting in Underestimation of Left Ventricular Hypertrophy"

_ijerph, 2021, doi:10.3390/ijerph182412867_

Round 1
Reviewer 1 Report
This is a simple and effective study, showing the effect of low QRS voltage on the diagnostic yeld of ECG in the detection of heart abnormalities.
I have two comments:
1) Would it be possible to estimate the diagnostic yeld of the Romhilt-Estes or Perugia scores in your population? Of course this would be an exploratory analysis only, as it would be post-hoc, but still it would be of great interest
2) in table 2 I recommend to add absolute numbers and 95% ICs for all the results. Of course you should declare the statistical method used to estimate ICs in the statistical analysis paragraph of the manuscript.
Reviewer 2 Report
The authors addressed the prevalence and determinants of low QRS voltages in the limb leads in a consecutive series of patients undergoing preoperative cardiac evaluation over 60 year-old. The main findings were that: 1) LQRSV in limb leads was not very rare (3%); 2) only 19% had an underlying heart disease or classic etiologies of LQRSV; 3) patients with "unexplained" LQRSV were more often male, had an increased myocardial mass and reduced LV dimensions and showed lower amplitude of Tallest R-vave an deepest S-wave in the precordial leads; 4) criteria for LV hypertrophy based on precordial QRS showed very low diagnostic accurancy in patients with unexplained LQRSV. This study is interesting as it provides some interesting new data on this poorly understood ECG parameter.
I have some comments:
1) two hospitals participated in the study: however, the vast majority of patients (98%) belonged to a single hospital. What is the reason for including the second center, as the number of cases that it could provide was so low?
2) the authors found a much higher prevalence of LQRSV evaluated in the second hospital. Despite the inhomogeneous sample sizes, the difference remained highly significant (p<0.001). I believe that the authors should try to elucidate why. In my opinion, there may be two different causes:
- A) technical, i.e. the way the ECG is acquired: e.g. patches at the limbs root versus pliers in the wrists and ankles; low-pass filter setting...
- B) patients characteristics
3) in the method section, how the ECG was performed should be stated and how the filters were set, as they can impact on QRS amplitude. In my clinical practice, I often observed that ECG filters are overlooked and set random. If filters were not set the same in all ECG machines, they may explain some observed differences in the low QRS voltage prevalence;
4) please clarify better how the control population was chosen and if they were just consecutive patients or if any matching was performed.
5) the preoperative cardiac screening protocol should also be described in the methods. In particular, was echocardiography performed in all patients?
6) I think a multivariable analysis for predictors of "unexplained" low QRS, apart for the univariate analysis showed in Table 1, should be performed;
7) in your discussion about the difference between limb and precordial QRS voltages, you should discuss that limb leads are bipolar and precordial leads are unipolar.
Round 2
Reviewer 2 Report
The revision has improved the manuscript and I have now understood the methods. Two minor comments:
-In the methods section the sentence "Patients routinely underwent ECG, echocardiography, chest radiography, and pulmonary function test within a week." is a repetition of the previous one and can be omitted
- I would specify why you did not simply compare the 292 patients with LQRSV of hospital 1 versus the 9537 patients without LQRSV of hospital 1. Probably it is because of missing data requiring to select a smaller control sample with all data collected.
